# STRUCTURED CONTRASTIVE LEARNING FOR INTERPRETABLE LATENT REPRESENTATIONS

## ABSTRACT

Neural networks exhibit severe brittleness to semantically irrelevant transformations. A mere 75ms electrocardiogram (ECG) phase shift degrades latent cosine similarity from 1.0 to 0.2, while sensor rotations collapse activity recognition performance with inertial measurement units (IMUs). We identify the root cause as "laissez-faire" representation learning, where latent spaces evolve unconstrained provided task performance is satisfied. We propose **Structured Contrastive Learning (SCL)**, a framework that partitions latent space representations into three semantic groups: *invariant features* that remain consistent under given transformations (e.g., phase shifts or rotations), *variant features* that actively differentiate transformations via a novel variant mechanism, and *free features* that preserve task flexibility. This creates controllable push-pull dynamics where different latent dimensions serve distinct, interpretable purposes. The variant mechanism enhances contrastive learning by encouraging variant features to differentiate within positive pairs, enabling simultaneous robustness and interpretability. Our approach requires no architectural modifications and integrates seamlessly into existing training pipelines. Experiments on ECG phase invariance and IMU rotation robustness demonstrate superior performance: ECG similarity improves from 0.25 to 0.91 under phase shifts, while WISDM activity recognition achieves 86.65% accuracy with 95.38% rotation consistency, consistently outperforming traditional data augmentation. This work represents a paradigm shift from reactive data augmentation to proactive structural learning, enabling interpretable latent representations in neural networks.

## 1 INTRODUCTION

Neural networks often exhibit unexpected sensitivity to transformations that should be semantically irrelevant. This transformation brittleness manifests across domains: image classification systems fail under minor rotations (Gao et al., 2020), signal processing models become unreliable with temporal shifts (Volpi & Murino, 2019), and activity recognition systems collapse under sensor orientation changes.

As an example, this vulnerability became stark in similar ECG signal retrieval for cardiac disease detection. Retrieval based on variational autoencoders (VAEs) failed catastrophically when identical cardiac waveforms were temporally shifted (Shen et al., 2025). As shown in Figure 1, even minor temporal shifts lead to significantly different latent representations, despite identical signal morphology. Cosine similarity drops rapidly from 1.0 to below 0.2, indicating that the learned representations unsuitable for clinical similarity retrieval.

This brittleness stems from *"laissez-faire" representation learning*—where learned representations evolve unconstrained as long as task performance is satisfied (Zhao et al., 2022). Traditional training objectives optimize solely for task performance while providing no explicit guidance on inter-sample relationships across transformations (Higgins et al., 2018). Information-theoretic analysis reveals that standard loss functions maximize mutual information between inputs and task-relevant features while remaining agnostic to semantic structure preservation (Tian et al., 2020). Networks can become excessively invariant to meaningful changes while remaining overly sensitive to irrelevant ones (Jacobsen et al., 2018), with fundamental impossibility results proving that invariance alone cannot identify meaningful latent structure (Bing et al., 2023).

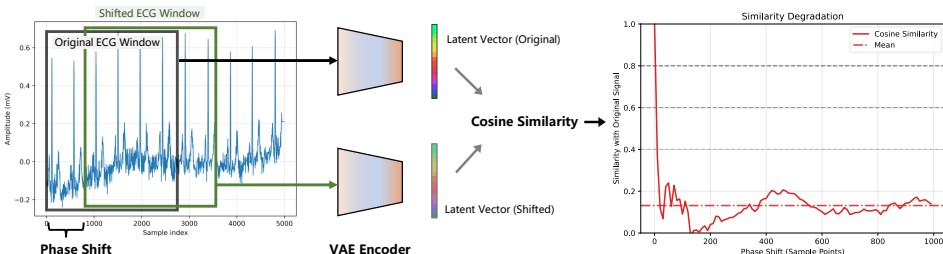

Figure 1: Transformation brittleness in latent representations. Traditional VAEs exhibit severe phase sensitivity: identical ECG waveforms at different temporal positions produce dramatically different latent vectors, with cosine similarity degrading from 1.0 to below 0.2 across phase shifts (sampling rate: 400 Hz).

The standard response has been data augmentation—exposing models to transformed training data. Contrastive methods like SimCLR (Chen et al., 2020) and MoCo (He et al., 2020) achieve success by maximizing agreement between augmented views while minimizing agreement with other samples. However, these approaches typically apply uniform constraints across all latent dimensions and operate through discrete sampling of transformation space, providing no explicit control over representation structure.

Recent advances identify critical limitations: forcing invariance to all augmentations can be suboptimal for downstream tasks (Xiao et al., 2020), with inherent trade-offs between accuracy and invariance that cannot be resolved through augmentation alone (Zhao et al., 2022). Data augmentation addresses symptoms rather than causes—increasing exposure to transformations without providing principled control over representation organization.

We propose a paradigm shift from discrete data augmentation to structured latent space learning through explicit feature partitioning and contrastive control. Our framework transforms neural networks from black boxes into interpretable glass box systems with controllable semantic organization. The core innovation lies in partitioning latent representations into functionally distinct groups: **invariant features** that remain consistent across transformations, **variant features** that actively differentiate between transformations, and **free features** that remain unconstrained for task-specific optimization.

Central to this structured contrastive learning is a contrastive objective that simultaneously pulls invariant features together while pushing features of different samples apart, enhanced through a *variant mechanism* that encourages variant features to differentiate even within positive pairs. This creates sophisticated learning where different latent aspects serve distinct, controllable purposes. Our approach builds upon disentanglement frameworks (Higgins et al., 2018), invariant causal representation learning (Mitrovic et al., 2020), and extends contrastive learning approaches like ContrastVAE (Wang et al., 2022) and multi-level feature learning (Xu et al., 2022), while addressing fundamental limitations in purely invariance-based methods (Bing et al., 2023; Zhao et al., 2022).

Our framework provides key advances: fine-grained control over different representation aspects through explicit partitioning, enhanced interpretability while maintaining task accuracy, and seamless integration into existing training pipelines without architectural modifications. We demonstrate effectiveness through comprehensive experiments on ECG phase invariance (improving cosine similarity from 0.25 to 0.91) and IMU rotation robustness (achieving 86.65% accuracy with 95.38% rotation consistency).

Our contributions represent a paradigm shift from reactive data augmentation to proactive structural learning: (1) systematic identification of transformation brittleness as a universal neural network limitation; (2) a structured latent space learning framework transforming black box networks into interpretable glass box systems; (3) a structured contrastive learning mechanism with variant-enhanced control providing fine-grained feature organization; and (4) a practical training methodology enhancing interpretability and robustness without sacrificing performance. This opens new possibilities for deploying deep learning systems in critical applications requiring both performance and interpretability.

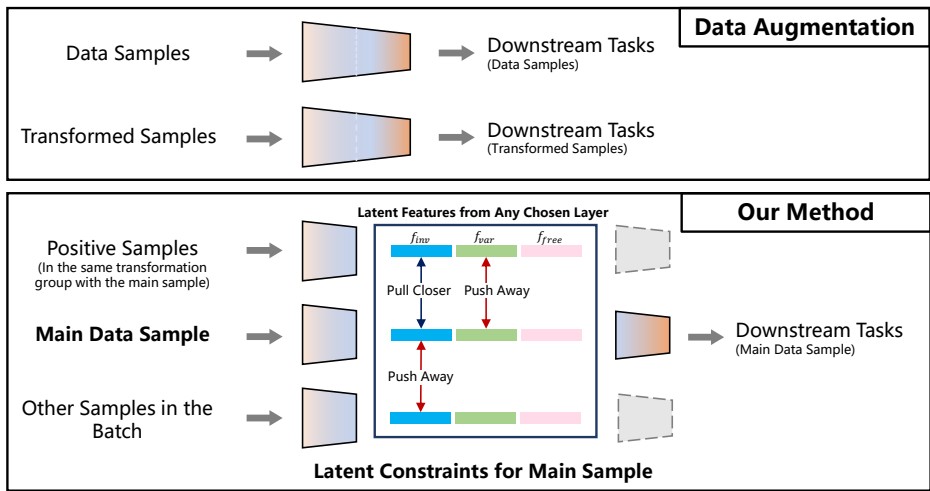

Figure 2: **Structured contrastive learning in latent space. Top:** Traditional data augmentation provides no control over latent representations. **Bottom:** Our method partitions features into invariant (pulled together), variant (pushed apart via variant mechanism), and free (unconstrained) components, transforming neural networks into interpretable systems with controllable semantic meaning.

## 2 METHOD

We present our structured contrastive learning framework that creates interpretable latent representations through explicit feature partitioning and controllable contrastive objectives.

### 2.1 STRUCTURED LATENT SPACE PARTITIONING

Traditional neural networks learn task-specific representations through purely task-oriented objectives:

$$\mathcal{L}_{\text{traditional}} = \mathcal{L}_{\text{task}}(f(x), y) \tag{1}$$

where $f(\cdot)$ is the encoder function and $y$ is the task label. This allows latent representations to evolve freely, causing semantically similar inputs $x$ and $T(x)$ (where $T$ represents transformations like phase shifts or rotations) to produce arbitrarily different representations: $||f(x) - f(T(x))||$ can be large despite semantic equivalence.

Our key insight is to explicitly model inter-sample relationships through structured contrastive learning:

$$\mathcal{L}_{\text{structured}} = \mathcal{L}_{\text{task}} + \lambda \mathcal{L}_{\text{contrastive}} \tag{2}$$

We partition the latent representation $f(x) \in \mathbb{R}^d$ into three semantically distinct subgroups:

$$f(x) = [f_{\text{inv}}(x), f_{\text{var}}(x), f_{\text{free}}(x)] \tag{3}$$

where $d_{\text{inv}} + d_{\text{var}} + d_{\text{free}} = d$. Each subgroup serves a distinct purpose: **Invariant features** ($f_{\text{inv}}$) encode task-relevant information that should remain consistent across transformations; **Variant features** ($f_{\text{var}}$) capture transformation-specific information that should change predictably; **Free features** ($f_{\text{free}}$) learn unconstrained representations, preserving network flexibility.

### 2.2 VARIANT-ENHANCED STRUCTURED CONTRASTIVE LEARNING

Our central innovation lies in the variant mechanism, which creates a sophisticated push-pull dynamic. For positive pairs $(x, T(x))$ where $T$ represents semantic-preserving transformations, our framework enforces:

- **Invariant constraint**: $\mathcal{D}(f_{\text{inv}}(x), f_{\text{inv}}(T(x))) \to 0$
- **Variant constraint**: $\mathcal{D}(f_{\text{var}}(x), f_{\text{var}}(T(x))) \to \max$

where $\mathcal{D}(\cdot, \cdot) = 1 - \cos(\cdot, \cdot)$ measures the cosine distance. This dual objective ensures that invariant features capture transformation-agnostic semantics while variant features encode transformation-specific information.

Our structured contrastive loss formalizes this dual objective as:

$$\mathcal{L}_{\text{contrastive}} = \frac{\mathcal{D}(f_{\text{inv}}(x), f_{\text{inv}}(T(x)))}{[1 + \beta \cdot \mathcal{D}(f_{\text{var}}(x), f_{\text{var}}(T(x)))] \cdot \mathcal{D}(f_{\text{inv}}(x), f_{\text{inv}}(x_{\text{neg}}))} \tag{4}$$

where $\beta \geq 0$ controls variant feature differentiation strength. The variant mechanism creates elegant dynamics: invariant features of positive pairs are pulled together (numerator), while variant features are pushed apart via the beta term in the denominator. When variant features are too similar, the denominator increases, making the loss larger and encouraging differentiation. The denominator also ensures negative samples remain distant in the invariant subspace. When $\beta = 0$, we recover standard contrastive learning; when $\beta > 0$, we actively encourage variant features to encode transformation differences.

### 2.3 COMPLETE FRAMEWORK INTEGRATION

Our complete objective function seamlessly integrates task performance with structured representation learning:

$$\mathcal{L}_{\text{total}} = \mathcal{L}_{\text{task}}(h(f(x)), y) + \lambda \mathcal{L}_{\text{contrastive}}(f(x), f(T(x)), f(x_{\text{neg}})) \tag{5}$$

where $h(\cdot)$ is the task-specific head and $\lambda$ balances task performance with structural constraints. A key strength lies in non-invasive integration: our method requires no architectural modifications and works with any intermediate layer representation $f(x)$.

This framework fundamentally transforms neural networks from black boxes—where latent representations evolve unpredictably—into interpretable systems where different feature groups have explicit, controllable semantic meanings. By explicitly decoupling functional aspects (e.g., separating what an activity is from how it's oriented), practitioners gain unprecedented interpretability and control over model behavior while maintaining task performance.

## 3 EXPERIMENTS AND RESULTS

We validate our structured latent space learning framework through two complementary experimental settings that demonstrate universal applicability across different neural architectures, data modalities, transformation types, and task objectives. **ECG phase invariance** (similarity improved from 0.25 to 0.91) demonstrates transformation from phase-sensitive to morphology-focused medical retrieval, while **IMU rotation robustness** (86.65% accuracy with 95.38% rotation consistency) shows structured learning's effectiveness in discriminative classification tasks.

### 3.1 ECG PHASE INVARIANCE IN MEDICAL SIGNAL RETRIEVAL

Medical signal retrieval requires robust similarity matching focusing on morphological patterns rather than temporal alignment. Traditional VAEs exhibit severe phase sensitivity, causing identical ECG waveforms shifted by milliseconds to produce drastically different latent representations.

**Experimental Setup.** We employ a residual VAE architecture with four 1D convolutional blocks (32, 64, 128, 256 channels) trained on three ECG datasets: SaMi-Trop (Cardoso et al., 2016), PTB-XL (Wagner et al., 2020), and CODE-15% (Ribeiro et al., 2020). All signals undergo standardized preprocessing to 400 Hz, 7-second duration with NeuroKit2 filtering. The VAE learns 256-dimensional latent representations, with our structured approach treating all dimensions as invariant features to encourage morphological consistency across phase shifts.

We compare three approaches: **Baseline** uses the pre-trained VAE without modification; **Data Augmentation** fine-tunes the pre-trained VAE on phase-shifted versions of the original data; **Our**

**Method** fine-tunes the pre-trained VAE with structured contrastive learning to enforce phase invariance in the latent space. For positive pair generation, we create multiple random phase shifts for each ECG signal, then randomly select one as the anchor (main sample) and others as positive samples, preventing the model from learning only specific phase shift patterns.

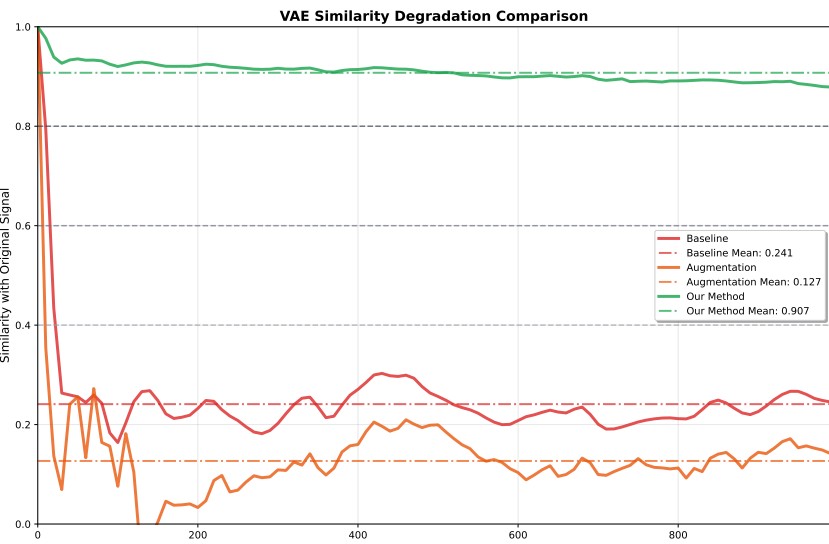

Figure 3: Phase invariance transformation results.

**Phase Invariance Results.** Figure 3 reveals our method's dramatic transformation effectiveness. **Critically, data augmentation performs worse than the baseline**—a counterintuitive finding that exposes a fundamental limitation: simply exposing models to augmented data without explicit structural guidance can lead to overfitting on augmentation patterns rather than learning true invariance. Our structured approach maintains remarkably stable similarity (0.907) by learning continuous invariance manifolds rather than discrete transformation instances.

**Clinical Retrieval Effectiveness.** Figure 4 demonstrates practical clinical impact. Both baseline and data augmentation methods are "phase-locked"—retrieving signals matching not only morphological patterns but also specific phase alignment and cardiac timing, which is likely due to the MSE reconstruction loss during training. Our method successfully retrieves morphologically similar signals with clear phase misalignment and different cardiac periods, achieving high similarity scores. This phase-agnostic retrieval is essential for clinical applications where pathological patterns appear at different temporal positions.

## 3.2 IMU ROTATION INVARIANCE IN ACTIVITY RECOGNITION

Human activity recognition faces the challenge that sensor orientation dramatically affects representations, causing identical activities to appear different under orientation changes. **Our structured learning achieves 86.65% accuracy with 95.38% rotation consistency**, demonstrating how explicit feature decoupling creates robust classifiers that understand *what* activity is performed independently of *how* the sensor is oriented.

**Experimental Setup.** We use the WISDM dataset (Kwapisz et al., 2011) with 6 activities from smartphone accelerometers. Our 1D CNN employs three convolutional blocks (64, 128, 256 channels) projecting to 128-dimensional features. We test two partitioning strategies: **invariant + free** features and **invariant + variant** features with the variant mechanism. Data augmentation applies random 3D rotations, creating positive pairs from the same activity window and negative pairs from different windows.

**The variant mechanism's structured feature decoupling** demonstrates clear advantages in Table 1: our structured method outperforms even standard contrastive learning with both highest ac-

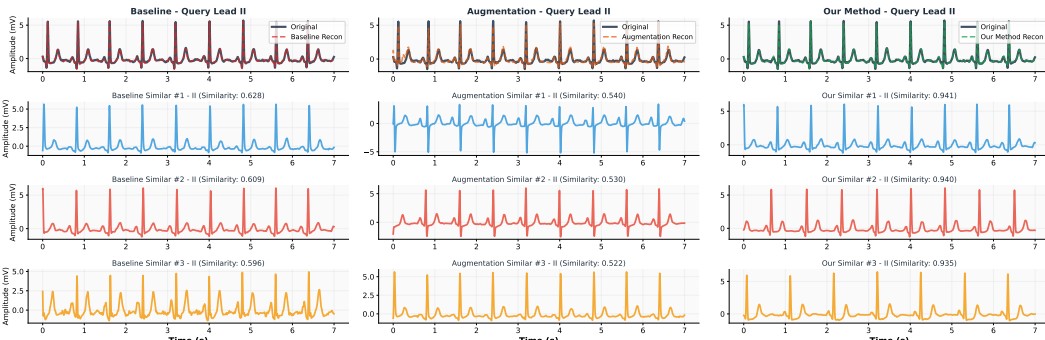

Figure 4: Clinical query effectiveness demonstration. Our method (rightmost) successfully retrieves morphologically similar signals regardless of phase alignment (0.935-0.941 similarity), while baseline and data augmentation methods remain "phase-locked," achieving much lower similarities (0.522-0.628) and missing clinically relevant patterns.

Table 1: Performance comparison: classification accuracy and rotation robustness. Rotation Consistency measures the percentage of test samples where the model produces identical predictions before and after applying 3D rotations, quantifying robustness to sensor orientation changes. Results for 32 invariant dimensions: Standard Contrastive uses invariant + free features, while Structured Contrastive uses invariant + variant features with variant mechanism. Our structured approach achieves highest performance while maintaining excellent robustness.

| Method | Accuracy (%) | Rotation Consistency (%) |
|---|---|---|
| Baseline | 58.64 | 64.97 |
| Data Augmentation | 84.03 | 94.57 |
| Standard Contrastive | 84.90 | 95.11 |
| Structured Contrastive | **86.65** | **95.38** |

curacy (86.65%) and excellent rotation consistency (95.38%), validating that explicit feature partitioning provides benefits beyond simple invariance learning.

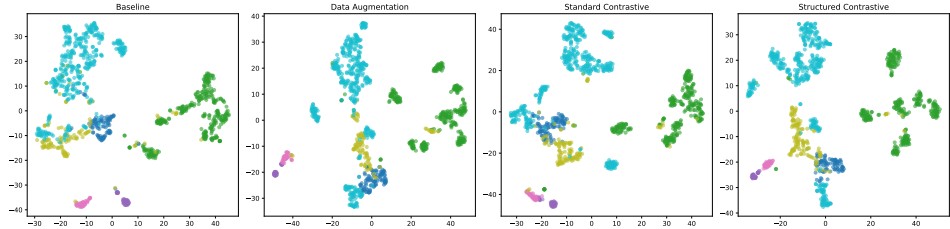

Figure 5: Feature space organization through structured learning. t-SNE visualizations show a progressive organization from baseline scatter (leftmost) to our structured clusters (rightmost) visually demonstrates the structured transformation—converting chaotic latent spaces into interpretable, organized representations with clear activity separation.

**Glass Box Transformation Visualization.** Figure 5 provides compelling visual evidence of our method's glass box transformation. The progression shows increasingly organized representations, with our structured approach achieving the most clearly separated activity clusters. Beyond inter-class separation, **our structured contrastive method also enhances intra-class distinction**—within each activity cluster, individual samples maintain more distinct positions rather than collapsing into tight, indistinguishable groups. This dual organizational benefit demonstrates how structured learning transforms unpredictable latent spaces into interpretable, controllable representations that preserve both semantic clustering and individual sample identity.

Table 2: Ablation study: feature dimension allocation impact. The accurency was evaluated under isolated rotations along the X, Y, and Z axes, and a combined transformation. Systematic evaluation reveals optimal configurations for different partitioning strategies. The variant mechanism consistently outperforms standard contrastive learning across all configurations, with 32 invariant dimensions achieving optimal performance.

| Method | Invariant Dims | Axis-Specific Accuracy (%) | | | |
|---|---|---|---|---|---|
| | | X-Axis | Y-Axis | Z-Axis | Combined |
| **Standard Contrastive** | 0 (AUG) | 83.70 | 84.74 | 83.43 | 84.25 |
| | 32 | 82.99 | **86.35** | 84.43 | **85.84** |
| | 64 | **84.59** | 85.72 | **84.62** | 85.09 |
| | 96 | 83.50 | 85.57 | 83.52 | 84.57 |
| | 128 | 83.51 | 85.22 | 83.35 | 84.41 |
| **Structured Contrastive** | 0 | 85.07 | 87.64 | 84.78 | 87.14 |
| | 32 | 85.74 | 88.19 | 84.97 | 87.69 |
| | 64 | 84.97 | 87.24 | 85.19 | 86.68 |
| | 96 | 84.07 | 86.30 | 84.95 | 85.70 |
| **Baseline** | 0 | 39.77 | 66.52 | 67.83 | 60.42 |

Table 2 presents a systematic analysis of how different invariant dimension allocations affect performance. The variant mechanism demonstrates consistent superiority across all configurations, with structured contrastive outperforming standard contrastive by 2-3% in most settings. The 32-invariant dimension configuration emerges as optimal (87.69% combined accuracy), suggesting that moderate feature partitioning strikes the best balance between invariance learning and discriminative capacity.

Notably, structured contrastive with 0 invariant dimensions achieves excellent performance (87.14%), nearly matching the optimal configuration. This seemingly paradoxical result reveals that the variant mechanism's explicit relationship modeling creates beneficial organization even without designated invariant features, indicating that structured learning benefits extend beyond simple feature partitioning. Standard contrastive learning shows a relatively flat performance curve (84.25-85.84% range), while structured contrastive exhibits clearer sensitivity to dimension allocation, suggesting more effective exploitation of feature partitioning advantages.

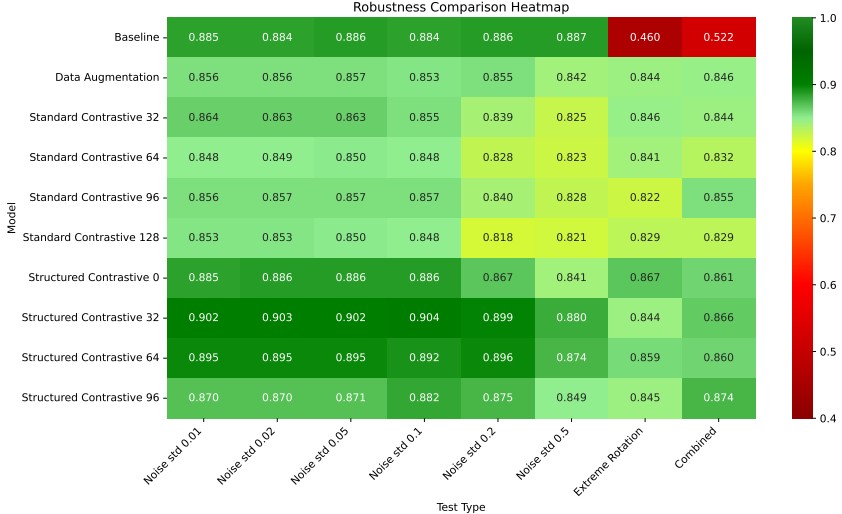

Figure 6: Stress test performance: noise, rotation, and combined challenges. Our structured approaches (bottom rows) maintain consistently high performance across all stress conditions, demonstrating the robustness benefits of structured feature organization. The variant mechanism shows particular resilience under extreme combined stress scenarios. Values in the heatmap indicate accuracy under each test condition.

**Comprehensive Stress Testing Analysis.** To validate the practical robustness of our glass box transformation, we conducted systematic stress tests across multiple perturbation types. The results in Figure 6 reveal distinct robustness profiles across learning approaches. The baseline model exhibits strong noise resistance (0.885-0.887) but catastrophic failure under rotation (0.460-0.522), revealing that untrained networks naturally handle additive noise but remain brittle to geometric transformations.

Data augmentation and standard contrastive methods show degraded noise performance compared to baseline (0.842-0.856 vs. 0.884-0.887), despite improving rotation robustness. This suggests these approaches sacrifice general robustness to achieve specific invariance, creating performance trade-offs rather than comprehensive improvement.

Our structured contrastive approaches achieve simultaneous excellence in both domains: strong noise resistance (0.880-0.904) while maintaining superior rotation robustness (0.844-0.867). This validates our theoretical prediction that variant features learn comprehensive transformation representations beyond rotation-specific patterns. The variant mechanism encourages variant features to capture the full spectrum of transformation-related information, including noise characteristics and geometric changes.

The 32-invariant dimension configuration consistently shows peak robustness across stress types, with even the "all-variant" configuration maintaining strong noise performance (0.841-0.886). This demonstrates that **the variant mechanism's explicit relationship modeling creates beneficial organization that transcends simple invariant/variant distinctions**. The comprehensive robustness analysis provides compelling evidence that structured learning creates fundamentally more robust neural representations through principled feature organization, where different feature groups can specialize in handling different perturbation types simultaneously.

## 4 ANALYSIS AND DISCUSSION

Our experimental results reveal several key insights about structured learning's advantages over traditional approaches.

A striking finding is that structured contrastive learning consistently outperforms data augmentation across both domains. In ECG experiments, data augmentation actually degraded performance compared to baseline, while our method achieved dramatic improvements. This counterintuitive result illuminates a fundamental limitation: data augmentation operates through discrete sampling of transformation space, potentially leading to overfitting on specific augmentation patterns rather than learning true invariance. Our structured approach learns continuous manifolds in latent space that capture transformation essence, enabling better generalization to unseen parameters.

The ablation studies demonstrate sophisticated benefits of explicit feature partitioning. Rather than constraining all features uniformly—which can suppress discriminative information—our method allows different feature groups to serve distinct semantic purposes. The variant mechanism actively encourages this decoupling by penalizing similar variant features between positive pairs, creating rich representational spaces that achieve simultaneous robustness and interpretability. Notably, the variant mechanism provides benefits even when all features are designated as "variant," suggesting that explicit relationship modeling helps create more organized representations regardless of partition specifics.

Success across radically different domains—medical signals and activity recognition, generative and discriminative models, temporal and spatial transformations—demonstrates universal applicability. This stems from addressing the fundamental issue of unconstrained latent space learning rather than architecture-specific problems. Our explicit feature partitioning provides unprecedented interpretability: practitioners can monitor feature group behavior, diagnose failures, control model behavior, and transfer insights across domains. This represents a paradigm shift from post-hoc interpretability to intrinsic interpretability designed from the ground up.

## 5 CONCLUSION

We present Structured Contrastive Learning, a unified framework that creates interpretable latent representations through explicit feature partitioning and contrastive objectives. Our approach addresses transformation brittleness—a universal problem where networks exhibit severe sensitivity to semantically irrelevant input changes.

The key contributions of this work include: (1) Systematic identification of transformation brittleness across neural architectures and applications; (2) A structured contrastive learning framework with explicit feature partitioning into invariant, variant, and free components; (3) The variant mechanism that actively encourages feature decoupling while maintaining task performance; (4) Superior performance compared to traditional data augmentation across medical signal analysis and activity recognition.

The core technical innovation lies in providing explicit structural guidance rather than relying on implicit learning through data exposure. Our method creates representations that are simultaneously more robust and interpretable than traditional approaches, requiring no architectural modifications while integrating seamlessly into existing training pipelines.

This work represents a paradigm shift toward controllable, interpretable neural networks. By providing explicit control over representation aspects, our approach enables reliable deployment in critical applications and better model understanding. Future directions include extension to multi-modal learning, few-shot learning applications, and hierarchical feature decoupling for complex transformations. Our framework effectively bridges the long-standing trade-off between performance and interpretability, paving the way for deep learning systems that are both powerful and understandable—a crucial step for their responsible use in real-world applications.

*Use of Large Language Models: We acknowledge the assistance of Claude Sonnet 4 (Anthropic) in improving the clarity and language quality of this manuscript.*

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
