# OpenReview forum: "Structured Contrastive Learning for Interpretable Latent Representations"
_ICLR.cc/2026/Conference — ICLR 2026 Conference Withdrawn Submission_

### Official Review · Reviewer_che7 · 2025-10-19

**Soundness:** 2
**Presentation:** 2
**Contribution:** 2
**Rating:** 4
**Confidence:** 4

**Summary:**

This paper introduces Structured Contrastive Learning (SCL), a framework designed to improve the robustness of neural network representations against irrelevant input transformations (e.g., phase shifts in ECGs, rotations in IMU data). SCL partitions latent representations into invariant, variant, and free features, leveraging a variant mechanism to enhance contrastive learning. Experiments are conducted on ECG similarity tasks and IMU-based activity recognition.

**Strengths:**

1. The three-part feature partitioning is well-motivated and directly addresses practical challenges, such as phase sensitivity in ECG analysis.
2. The paper is clearly and effectively written, making the proposed method accessible.
3. The experiments demonstrate the method’s advantages over the selected baselines (and datasets).

**Weaknesses:**

1. The paper overlooks key works in time series contrastive learning, such as T-Loss [1] and TS2Vec [2], which should be discussed and included as baselines for a comprehensive comparison.
2. The evaluation is limited to only two datasets, which is relatively narrow for a generalist conference like ICLR (see, for example, the experimental section of [1]).
3. The training process—specifically, how the task and contrastive losses are combined—lacks clarity, making reproducibility challenging.

[1] Unsupervised Scalable Representation Learning for Multivariate Time Series, Franceschi et al, NeurIPS 2019

[2] TS2Vec: Towards Universal Representation of Time Series, Yue et al, AAAI 2022

**Questions:**

1. Why were T-Loss, TS2Vec, and other established baselines from this literature not considered in the experiments?
2. Could you extend the experiments to include more datasets or tasks to validate the method’s generalizability further?
3. How are the task and contrastive losses optimized together? Is it done jointly, alternately, or through another strategy? Clarifying this would improve reproducibility.

---

### Official Review · Reviewer_BBkR · 2025-10-29

**Soundness:** 3
**Presentation:** 2
**Contribution:** 2
**Rating:** 2
**Confidence:** 4

**Summary:**

This paper introduces Structured Contrastive Learning (SCL), a framework that partitions latent space representations into three semantic groups: invariant features that remain consistent under given transformations (e.g., phase shifts or rotations), variant features that actively differentiate transformations via a novel variant mechanism, and free features that preserve task flexibility. This creates controllable push-pull dynamics where different latent dimensions serve distinct, interpretable purposes.

**Strengths:**

1 This paper is well written and organized.
2 Extensive experiments are conducted.

**Weaknesses:**

1.	The authors' motivation to shift "from data augmentation to structured contrastive learning" relies primarily on empirical observations (such as ECG similarity decline), but lacks formal theoretical analysis or addresses unresolved gaps in the literature. For example, "laissez-faire representation learning" is described as the root cause, but the concept is neither formalized nor quantified, nor is its universality across different tasks verified. Therefore, the motivation section lacks rigorous theoretical support and quantitative evidence.
2.	The article's contributions may be exaggerated. The authors repeatedly emphasize "interpretable latent space" and "glass box transformation," but fail to provide interpretability assessment metrics or experiments. Claiming that "enhanced interpretability" requires specific interpretability metrics or user-friendly visualization experiments, rather than simply relying on t-SNE clustering diagrams.
3.	The comparison only includes SimCLR and MoCo-style contrastive learning and data augmentation, but does not compare recent structured/decoupling methods. The authors claim to preserve both semantic clustering and individual sample identity, yet they do not provide any cross-domain evaluations, such as cross-subject, cross-sensor, or cross-dataset experiments, to substantiate this claim.
4.	In the ECG experiment, the authors claimed that data augmentation “actually degraded” (Fig. 3), but did not explain the augmentation parameter space and training strategy, which may have hyperparameter mismatch or undertraining problems.
5.	Eq.4 loss appears incompatible with standard contrastive learning objectives, lacking a derivation of its relationship with existing objectives. Compared to labeled contrastive losses (such as InfoNCE), its structure is clearly unstable, lacking a temperature term or probabilistic meaning, and prone to gradient explosion/vanishing.
6.	The paper assumes f(x) = [f_inv, f_var, f_free], but does not explicitly explain how these three parts are allocated in terms of dimensions or how independent optimization is achieved through gradient isolation. "Semantic decoupling" may only be a formal division, lacking a true guarantee of separability.

**Questions:**

1.	The definition of the "rotation consistency" metric is vague and does not explain how to judge "prediction consistency"？

---

### Official Review · Reviewer_MbiW · 2025-10-30

**Soundness:** 2
**Presentation:** 2
**Contribution:** 1
**Rating:** 2
**Confidence:** 4

**Summary:**

The paper investigated the shift-invariancy in signals for deep learning. The authors propose an approach, SCL, to partition the latent space into semantic groups and apply contrastive learning inside these groups.

**Strengths:**

The shift-invariancy is an important problem for most of the temporal signals. Investigating this problem in self-supervised learning for temporal signals is novel.

**Weaknesses:**

The paper lacks a thorough literature review. The references are minimal and do not include key related work. In particular, the authors should compare their approach with prior studies such as [1], which explore similar problems in supervised learning.

The proposed method also shows limited novelty. Related works like [2] have already introduced the idea of separating latent spaces or embedding representations to capture invariant and variant factors. The paper does not clearly state how its contribution differs from or extends such prior approaches. Clarifying the conceptual or technical distinction would significantly strengthen the paper.

The experimental evaluation is narrow. The results are based only on ECG data, which makes it difficult to assess the generality of the method. Since the framework is meant to handle temporal signals, I recommend including at least 3–4 additional modalities (e.g., EEG, PPG, EMG) to provide a more comprehensive validation.

[1] Shifting the Paradigm: A Diffeomorphism Between Time Series Data Manifolds for Achieving Shift-Invariancy in Deep Learning. ICLR 2025.
[2] What Should Not Be Contrastive in Contrastive Learning. ICLR 2021.

**Questions:**

1) How does your method conceptually differ from [1] and [2]?

2) What aspects of your latent space design are novel compared to the multi-space representation strategy in [2]?

3) What are the main challenges that prevent extending the experiments to other temporal signals?

---

### Note · Authors · 2025-11-19

**Comment:**

We thank the reviewers for their valuable feedback. We have decided to withdraw the submission after careful consideration.

**Withdrawal Confirmation:**

I have read and agree with the venue's withdrawal policy on behalf of myself and my co-authors.